# Prevalence of Septate Uterus in a Large Population of Women of Reproductive Age: Comparison of ASRM 2016 and 2021, ESHRE/ESGE, and CUME Diagnostic Criteria: A Prospective Study

**DOI:** 10.3390/diagnostics14182019

**Published:** 2024-09-12

**Authors:** Isabel Carriles, Isabel Brotons, Tania Errasti, Alvaro Ruiz-Zambrana, Artur Ludwin, Juan Luis Alcazar

**Affiliations:** 1Department of Obstetrics and Gynecology, Clínica Universidad de Navarra, 31008 Pamplona, Spain; icarriles@unav.es (I.C.); ibrotons@unav.es (I.B.);; 2Department of Obstetrics and Gynecology, Medical University of Warsaw, 02-091 Warszawa, Poland; ludwin@cm-uj.krakow.pl

**Keywords:** septate uterus, diagnosis, ultrasound

## Abstract

In this study, we aimed to assess and compare the prevalence of septate uterus using the diagnostic criteria of the ESHRE-ESGE, ASRM 2016, ASRM 2021, and CUME classifications. This prospective observational study included 977 women of reproductive age. Each participant underwent a transvaginal ultrasound, and a 3D volume of the uterus was obtained for further analysis. Offline assessment of the uterine coronal plane was conducted to measure uterine wall thickness, fundal indentation length, and indentation angle. The diagnosis of a septate uterus was determined according to the criteria of the ESHRE-ESGE, ASRM, and CUME classifications. The prevalence of septate uterus was then calculated and compared across these classifications. The ESHRE-ESGE classification identified 132 women (13.5%) with a septate uterus. The 2016 ASRM classification identified nine women (0.9%), with an additional nine women falling into a grey zone. The 2021 ASRM classification identified fourteen women (1.4%), with eleven women in the grey zone. The CUME classification identified 23 women (2.4%). The prevalence of septate uterus was significantly higher when using the ESHRE-ESGE criteria compared to the 2016 ASRM [relative risk (RR): 7.33 (95% CI: 4.52–11.90)], the 2021 ASRM [RR: 5.28 (95% CI: 3.47–8.02)], and the CUME [RR: 5.94 (95% CI: 3.72–8.86)] (*p* < 0.001). Our findings indicate that the ESHRE-ESGE criteria result in a significantly higher prevalence of septate uterus compared to the ASRM and CUME criteria. The ASRM 2016 criteria may underdiagnose more than half of the cases.

## 1. Introduction

Congenital uterine anomalies (CUA) are anatomical abnormalities resulting from the embryological maldevelopment of the Müllerian ducts [1,2]. The optimal diagnostic tests for CUA include three-dimensional transvaginal ultrasound, laparoscopy or laparotomy performed in conjunction with hysteroscopy or hysterosalpingography (HSG), magnetic resonance imaging (MRI), and saline sonohysterography [3].

Several classification systems for congenital uterine anomalies have been proposed [4,5,6,7,8]. The two most commonly used in clinical practice are the American Society for Reproductive Medicine (ASRM) classification [4] and the joint classification by the European Society for Embryology and Human Reproduction (ESHRE) and the European Society for Endoscopic Gynecology (ESGE) [5].

In 2016, the ASRM classification was revised to consider a uterus as septate when there is an indentation of the uterine cavity with a depth greater than 15 mm and an indentation angle of less than 90° [9]. This criterion was modified in 2021 [10], a change that has been criticized due to the lack of an apparent rationale [11]. Additionally, this classification can leave some cases unclassified if they do not meet both criteria (an indentation larger than 15 mm but an angle wider than 90°, or vice versa), placing them in a grey zone.

The ESHRE-ESGE classification defines a septate uterus based on the indentation-to-wall-thickness (I:WT) ratio [12]. Some authors argue that the ESHRE-ESGE classification may lead to overdiagnosis and potential overtreatment [13,14]. In 2018, the Congenital Uterine Malformation Expert Group (CUME) proposed a new definition of a septate uterus, initially based on the depth of indentation, angle of indentation, and (I:WT) ratio [15], which was later simplified to a single criterion based on indentation depth in the fundal cavity [16].

The actual prevalence of congenital uterine anomalies, particularly the septate uterus, is unknown due to the asymptomatic nature of many cases. A recent meta-analysis reported a prevalence of septate uterus of 2.3% in an unselected population, rising to 3.0–15.4% in high-risk populations [3]. This meta-analysis did not specify the classification used, but it can be assumed that the studies employed the 1988 ASRM classification, as the more recent classifications did not exist at the time. The prevalence of septate uterus may vary with different classification systems. Previous studies have reported a high prevalence using the ESHRE-ESGE classification [17,18,19,20].

To the best of our knowledge, only one previous study has compared the 2016 ESHRE-ESGE, 2016 ASRM, and 2019 CUME classifications in women attending a specialized reproductive medicine center [16]. However, no studies have compared the prevalence of septate uterus in women attending a non-specialized center using the 2016 ASRM, 2021 ASRM, 2016 ESHRE-ESGE, and 2019 CUME classifications. Additionally, data on how the presence of uterine fibroids or adenomyosis affects the diagnostic measurements for septate uterus are limited [18].

The primary aim of this study is (i) to compare the prevalence of septate uterus in a large series of women of reproductive age using the ESHRE-ESGE, ASRM (2016 and 2021), and CUME classification criteria and (ii) to compare the prevalence by these criteria with aggregated evidence on the prevalence of septate uterus before the launching of these criteria [3]. A secondary objective is to assess whether the presence of uterine fibroids, adenomyosis, or intrauterine devices (IUD) affects the diagnosis of septate uterus.

## 2. Materials and Methods

### 2.1. Study Design

This is a prospective observational cohort study comprising a consecutive series of 1000 non-pregnant premenopausal women. The study was reported in accordance with the Strengthening the Reporting of Observational Studies in Epidemiology (STROBE) statement [21].

### 2.2. Patients

Eligible subjects for this study included women who attended our institution (Clinica Universidad de Navarra, Pamplona, Spain) for routine gynecological check-ups or who presented with gynecological symptoms such as abnormal bleeding or pain between October 2019 and December 2020. Recruitment was conducted by the first author in the outpatient office through personal invitations following a brief explanation of the study’s nature and objectives. Only women who provided informed consent were enrolled in the study.

The inclusion criteria were as follows:women aged 18 to 45 years old;asymptomatic or complaining of pelvic pain, uterine bleeding, or vaginal discharge;no virgo intacta (for allowing transvaginal ultrasound);

The exclusion criteria were as follows:previous hysterectomy for any reason;past or current history of gynecological cancer;previous metroplasty with uterine septum resection;menopause established.

Demographic characteristics, including patient age, number of gestations, number of abortions, number of deliveries, and number of preterm deliveries, were recorded and provided into an Excel data sheet file (Microsoft, version 2021) with anonymization of the patients’ data.

### 2.3. Ultrasound Evaluation

After a clinical examination, all participants underwent transvaginal ultrasound using several machines (Voluson E10, E8, and S10, GE Healthcare, Zipf, Austria) equipped with 5–9 MHz endovaginal probes. The examinations were performed by several experienced gynecologists (JLA, IC, IB, TE, ARZ, MA, and BO), all of whom are well-trained in 2D and 3D ultrasound techniques.

Following a comprehensive ultrasound evaluation of the uterus and adnexa, a 3D volume of the uterus was acquired according to a standardized protocol [22] for further analysis using 4DView™ software, Version 18 (GE). Subsequently, a single trained examiner (IC) conducted an offline assessment of the uterine coronal plane to measure the uterine wall thickness, fundal indentation depth, and indentation angle. Special attention was given to drawing the inter-cornual–interostial line instead of the simple interostial line and measuring the indentation angle, following the recommendations of Ludwin and Martins [23] (Figure 1).

When the contour of the uterine cavity fundus was convex, the indentation depth was considered to be 0 mm, and the indentation angle was recorded as 180°. The indentation-to-wall-thickness ratio (I:WT) was also calculated.

The presence and location of uterine fibroids, adenomyosis, intrauterine devices (IUDs), or other types of uterine congenital anomalies were documented. For the purpose of analysis, fibroids larger than 2 cm that distorted the uterine cavity were considered significant. Adenomyosis was diagnosed in the presence of at least two sonographic features associated with this condition [24].

The diagnosis of a septate uterus was determined according to four different criteria (2016-ESHRE-ESGE, 2016-ASRM, 2021-ASRM, and 2019-CUME) [9,12,16] (Figure 2, Figure 3 and Figure 4). The 2016-ASRM classification defines a septate uterus as having a uterine cavity indentation depth >15 mm and an indentation angle <90° [9]. The 2021-ASRM criteria define a septate uterus as having a cavity indentation >10 mm and an indentation angle <90° [10]. The ESHRE-ESGE classification identifies a septate uterus based on an indentation-to-wall-thickness (I:WT) ratio >50% with an external fundal indentation <50% [12]. The CUME criteria specify an indentation in the fundal cavity >10 mm with an external serosal indentation <10 mm [16].

Inter-observer reliability for classifying the uterus as septate or non-septate was good to very good across all classifications [25]. Additionally, the reliability of the measurements, independent of the diagnosis, was shown to be high [16].

### 2.4. Statistical Analysis

Categorical variables (presence of fibroids >2 cm, presence of an IUD, presence of adenomyosis, and presence of a septate uterus according to each classification) are presented as numbers and percentages. Continuous variables (patient age, number of gestations, number of abortions, number of deliveries, number of preterm deliveries, day of the cycle when the ultrasound evaluation was performed, uterine wall thickness, fundal indentation depth, and indentation angle) are presented as the mean with standard deviation (SD) or median with interquartile range (IQR), depending on the data distribution. The Kolmogorov–Smirnov test was used to assess the distribution of continuous data. The prevalence of septate uterus was estimated according to each classification and compared using Fisher’s exact test. The statistical analysis was performed using SPSS version 20.0 (SPSS Inc., Chicago, IL, USA). A *p*-value < 0.05 was considered statistically significant for all analyses.

### 2.5. Ethical Approval

Ethical approval for the study was obtained from the Institutional Review Board of the University of Navarra (Approval No. 2020-04) prior to commencement. All participants provided informed consent after the nature and objectives of the study were fully explained to them.

## 3. Results

During the study period, 1000 women were recruited. Seven women who had previously undergone metroplasty were excluded from the analysis. Additionally, in 12 cases (1.2%, 12/993), the coronal plane could not be accurately assessed due to the presence of a fundal uterine fibroid (*n* = 8), an IUD (*n* = 2), or poor-quality imaging (*n* = 2). Furthermore, four women were diagnosed with other congenital uterine anomalies (two cases of didelphys/bicorporeal uterus, one unicornuate uterus, and one T-shaped/dysmorphic uterus). These patients were also excluded, as the measurements required for diagnosing a septate uterus could not be performed or other congenital uterine anomalies were identified.

Therefore, 977 women were ultimately included in the analysis. All women were premenopausal. The clinical characteristics of these patients are presented in Table 1.

The majority of women (73%) were asymptomatic at the time of ultrasound examination. Twenty-nine women (2.7%) had an IUD in place. Ninety-six women (9.8%) had uterine fibroids larger than 2 cm that distorted the uterine cavity, while 56 women (5.7%) were diagnosed with adenomyosis. As mentioned earlier, only eight women were excluded due to the presence of fibroids, and two due to the presence of an IUD. It was observed that the percentage of women with fibroids, adenomyosis, or IUDs in whom a uterine septum was diagnosed was low.

The mean day of the cycle when the ultrasound was performed was 14.8 (SD: 7.9), ranging from day 1 to 40.

The overall data regarding the measurements of indentation depth, indentation angle, uterine wall thickness, and I:W ratio are shown in Table 2 and Figure 5. One hundred and ninety-three women (19.7%) had an indentation of 0 mm.

According to the 2016-ESHRE-ESGE classification, 132 women were diagnosed with a septate uterus (prevalence: 13.5%, 95% CI: 11.5–15.8%), with six cases classified as complete septum and 126 as partial septum. According to the 2016-ASRM classification, nine women were identified as having a septate uterus (prevalence: 0.9%, 95% CI: 0.5–1.7%), with six of these considered a complete septum. Additionally, nine women (0.9%, 95% CI: 0.5–1.7%) could not be classified due to either an indentation depth of less than 15 mm combined with an indentation angle of less than 90° or an indentation depth greater than 15 mm with an indentation angle greater than 90° (referred to as the “grey zone”). According to the 2021-ASRM classification, fourteen women had a septate uterus (prevalence: 1.4%, 95% CI: 0.9–2.4%), with six considered a complete septum. Furthermore, eleven women (1.1%, 95% CI: 0.6–2.0%) could not be classified due to either an indentation depth of less than 10 mm with an indentation angle of less than 90° or an indentation depth greater than 10 mm with an indentation angle greater than 90° (“grey zone”). According to the 2019-CUME classification, 23 women were diagnosed with a septate uterus (prevalence: 2.4%, 95% CI: 1.6–3.5%), with six cases classified as a complete septum and 17 as a partial septum. Interestingly, the six cases classified as a complete septum were the same across all four classifications.

The prevalence of septate uterus was significantly higher when using the ESHRE-ESGE classification compared to the 2016-ASRM classification [relative risk (RR): 7.33 (95% CI: 4.52–11.90), considering grey-zone cases as anomalies and RR: 14.67 (95% CI: 7.51–28.64), (*p* < 0.001)], the 2021-ASRM classification [RR: 5.28 (95% CI: 3.47–8.02), considering grey-zone cases as anomalies and RR: 9.43 (95% CI: 5.47–16.24), (*p* < 0.001)], and the CUME classification [RR: 5.94 (95% CI: 3.72–8.86), (*p* < 0.001)]. The prevalence of septate uterus was lower when using the 2016-ASRM classification compared to the 2019-CUME classification [RR: 0.39 (95% CI: 0.18–0.84), *p* = 0.0163]. The prevalence of septate uterus, according to the 2019-CUME classification, was similar to that of the 2016-ASRM classification when grey-zone cases were considered anomalies (RR: 0.78, 95% CI: 0.42–1.44, *p* = 0.4312) and to the 2021-ASRM classification, whether grey-zone cases were considered anomalies (RR: 1.09, 95% CI: 0.62–1.90, *p* = 0.7701) or normal (RR: 0.61, 95% CI: 0.31–1.07, *p* = 0.1395).

The distribution of cases classified as septate by the ESHRE-ESGE classification compared with the CUME, 2016-ASRM, and 2021-ASRM classifications is presented in Table 3.

The distribution of cases considered as septate by the 2016-ASRM classification compared to the CUME and 2021-ASRM classifications is shown in Table 4.

The distribution of cases considered as septate by the 2021-ASRM classification compared to the CUME classification is shown in Table 5.

## 4. Discussion

### 4.1. Summary of Findings

We have observed that the prevalence of septate uterus in a non-selected population is significantly higher (by almost six-fold) when the ESHRE-ESGE criteria are used compared to the ASRM and CUME criteria. The frequency of cases where the presence of uterine fibroids, IUD, or poor-quality 3D volumes precludes a diagnosis is low, but it may occur in some cases.

### 4.2. Interpretation of Findings

Currently, three different criteria are used to diagnose a septate uterus, namely the ASRM, the ESHRE-ESGE, and the CUME [10,12,16]. Significant debate has arisen regarding whether the ESHRE-ESGE classification might over-diagnose septate uterus compared to the ASRM classification [2,14,26]. In fact, Knez and colleagues reported that up to 58% of cases considered an arcuate uterus under the former ASRM classification would be reclassified as a partial septate uterus using the ESHRE-ESGE classification [18]. Schöller and co-workers showed in a retrospective study involving 920 women who underwent surgical treatment following a diagnosis of a Müllerian anomaly that the prevalence of septate uterus using the ESHRE-ESGE classification was 29.0% [19]. However, Heinonen and colleagues, in a retrospective analysis of 621 women diagnosed with a congenital uterine anomaly (CUA), found that the rates of septate uterus were similar using the ESHRE-ESGE and ASRM classifications (49.1% versus 49.0%) [17]. Nevertheless, it is important to note that these authors included cases diagnosed using both optimal and non-optimal methods.

Two systematic reviews have shown that the prevalence of septate uterus, as defined by the ASRM classification in high-risk populations (infertility and/or recurrent miscarriage), is 2.0–5.3% [3,27]. The prevalence of septate uterus in an unselected population was 2.3% [3]. This figure is similar to ours (1.8%) if we consider septate and “grey zone” cases together.

Few studies have compared the prevalence of septate uterus in the same population according to the ASRM and ESHRE-ESGE criteria. Ludwin and Ludwin assessed the prevalence of septate uterus in a consecutive series of 261 women attending a private center specializing in the diagnosis and treatment of CUAs [28]. These authors found that the prevalence of septate uterus was significantly higher when the ESHRE-ESGE criteria were used (16.9%) compared to the 1988 ASRM criteria (6.1%). Ouyang and co-workers reported similar findings [20]. These authors retrospectively compared the prevalence of septate uterus in a series of 53,540 infertile women. They observed that the prevalence of septate uterus according to the ESHRE-ESGE criteria was higher (11.3%) than when the ASRM criteria were used (3.8%).

We have shown that the prevalence of septate uterus in a consecutive series of women of reproductive age is significantly higher when the 2016 ESHRE-ESGE criteria are used compared to the 2016-ASRM, 2021-ASRM, and 2019-CUME criteria (13.5%, 0.9%, 1.4%, and 2.4%, respectively). We also observed that some cases (0.9%) could not be classified when the ASRM classification was used. We consider these findings clinically relevant, especially given the ongoing debate regarding the potential benefits of uterine metroplasty (septum resection). Despite the fact that the only randomized controlled study to date did not observe any beneficial effect on reproductive outcomes from this intervention [29], several meta-analyses of observational studies have concluded that metroplasty might reduce the risk of spontaneous abortion, preterm delivery, and fetal malpresentation and improve live birth rate [30,31,32,33,34,35]. Moreover, surgery in asymptomatic women is debatable.

This discrepancy could be due to the definition of a uterine septum itself. We observed that the agreement for diagnosing a complete septate uterus using any classification is almost 100%. This is in line with previous studies [28,36]. However, in the case of a partial septate uterus, challenges to providing a diagnosis may arise, depending on the classification used. Recently, Russo and colleagues have highlighted this issue. They observed significant discrepancies between different classifications (2016 ESHRE-ESGE, 2021 ASRM, and 2019 CUME) when the fundal indentation length was between 5 and 10 mm [37]. In this specific context (small indentations), the role of hysteroscopy as a diagnostic method could be considered, as there is some evidence that ultrasound might underestimate the actual indentation length in these cases [38,39].

### 4.3. Strengths and Limitations

The main strength of our study is that, to the best of our knowledge, it is the first study comparing the prevalence of septate uterus in a large series of consecutive women attending a center not specialized in reproductive medicine according to the three currently proposed diagnostic criteria, including the newest ASRM classification. Certainly, our study may have a selection bias. And our sample might not be representative of the general population of women of reproductive age, as the mean age of our sample was relatively high (35.0 years old).

Additionally, the sample size, although not statistically estimated, is large, and the rate of excluded women was very low.

However, we acknowledge that our study also has limitations. The main limitation is that the data regarding reproductive outcomes were obtained retrospectively by reviewing patients’ clinical records. Moreover, many women had never attempted to become pregnant, and we could not assess whether infertility is related to the presence of a uterine septum.

### 4.4. Future Research Agenda

We believe our results highlight the need to reconsider the current diagnostic criteria for diagnosing a septate uterus. We agree with some authors that a consensus is needed [14]. The focus should likely be on the definition of a partial septate uterus.

In addition, in the analysis of the role of metroplasty in improving reproductive outcomes, additional parameters, such as the length or surface area of the septum [40] or endometrial cavity volume, should be considered.

## 5. Conclusions

The prevalence of septate uterus in our population is significantly higher when the ESHRE-ESGE criteria are used compared to the ASRM and CUME criteria. This observation might imply that a significant proportion of women could be diagnosed as having a septate uterus, when in fact, they would not be diagnosed with such a condition if another classification were used.

## Figures and Tables

**Figure 1 diagnostics-14-02019-f001:**
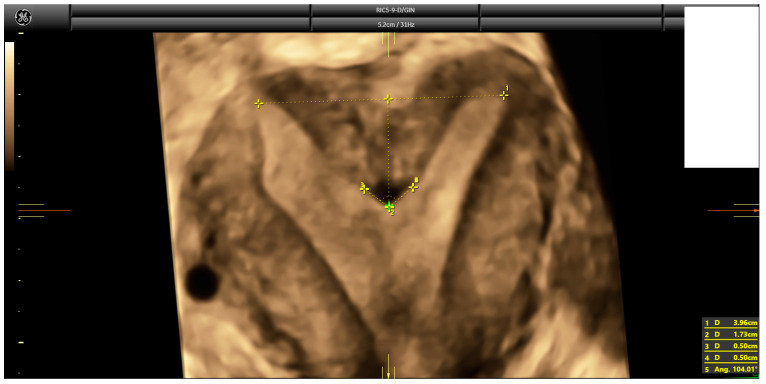
Three-dimensional ultrasound depicting how measurements were taken, according to Ludwin and Martin’s recommendations [23].

**Figure 2 diagnostics-14-02019-f002:**
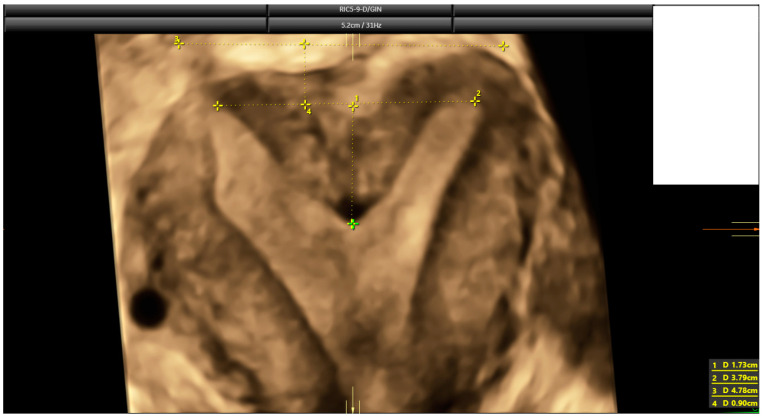
A case of septate uterus according to ESHRE-ESGE criteria. I:WT ratio is 192%.

**Figure 3 diagnostics-14-02019-f003:**
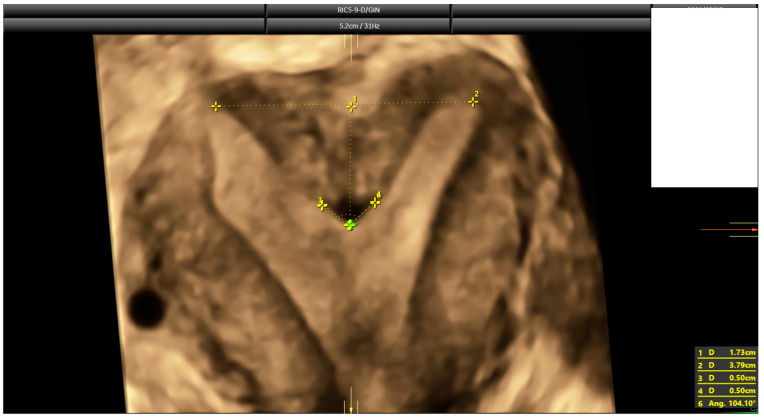
The same case as Figure 2. According to 2016 and 2021 ASRM criteria, this is a case that falls within the grey zone. Indentation length is 17.3 mm, but indentation angle is 104° (larger than 90°).

**Figure 4 diagnostics-14-02019-f004:**
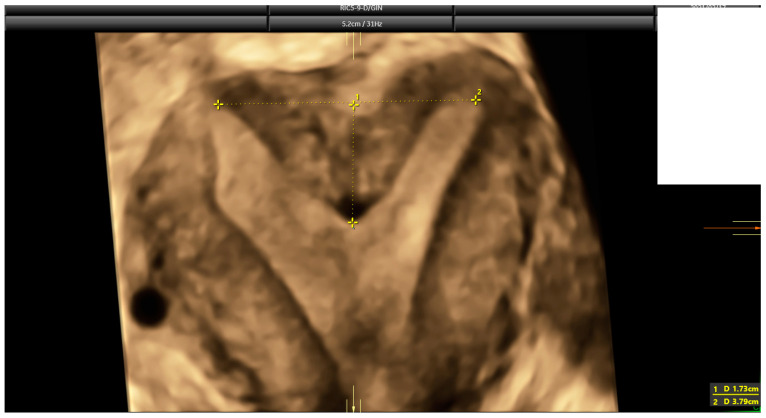
The same case as Figure 2 and Figure 3. According to CUME criteria, this is a septate uterus, with an indentation length of 17.3 mm.

**Figure 5 diagnostics-14-02019-f005:**
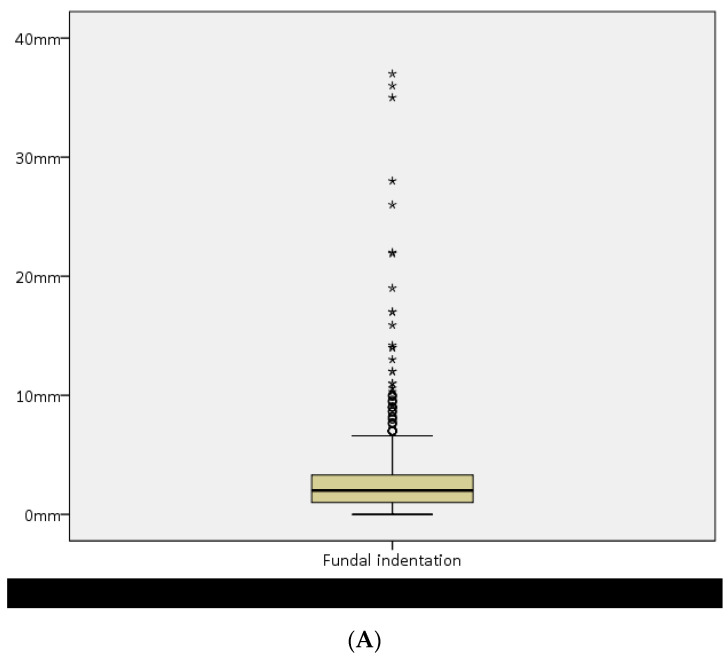
(**A**) Box plot showing median fundal indentation length, interquartile range, and outliers. (**B**) Box plot showing median indentation angle, interquartile range, and outliers. (**C**) Box plot showing median uterine wall thickness, interquartile range, and outliers. (**D**) Box plot showing median uterine wall thickness–indentation ratio (UWT:I), interquartile range, and outliers.

**Table 1 diagnostics-14-02019-t001:** Basic demographic characteristics of patients in the present study.

Age *	35.0 years (8.1)	Range: 18–45
Gravidity †	1.0 (2.0)	Range: 0–10
Parity †	0.0 (1.0)	Range: 0–6
Miscarriage †	0.0 (0.0)	Range: 0–6
Termination of pregnancy †	0.0 (0.0)	Range: 0–2

* Expressed as mean, standard deviation in parentheses. † Expressed as median, interquartile range in parentheses.

**Table 2 diagnostics-14-02019-t002:** Data regarding uterine measurements.

Parameter		
Indentation length *	2.0 mm (2.3)	Range: 0–37 mm
Uterine Wall thickness *	10.0 mm (3.7)	Range: 3.3–26.0 mm
Indentation angle *	169° (21.0)	Range: 0–180°
I:WT ratio *	20% (27.0)	Range: 1–596%

* Expressed as median, interquartile range in parentheses.

**Table 3 diagnostics-14-02019-t003:** Distribution of cases according to ESHRE-ESGE versus CUME, 2016-ASRM, and 2021-ASRM classifications.

	ESHER-ESGE Septate	ESHRE-ESGE No Septate	Total
CUME septate	23	0	23
CUME no septate	109	845	954
2016-ASRM septate	9	0	9
2016-ASRM gray zone	9	0	9
2016-ASRM no septate	114	845	959
2021-ASRM septate	14	0	14
2021-ASRM gray zone	11	0	11
2021-ASRM no septate	107	845	952

**Table 4 diagnostics-14-02019-t004:** Distribution of cases according to 2016-ASRM versus CUME and 2021-ASRM classifications.

	2016-ASRM Septate	2016-ASRM Grey Zone	2016-ASRM No Septate	Total
CUME septate	9	7	7	23
CUME no septate	0	2	952	954
2021-ASRM septate	9	5	0	14
2021-ASRM gray zone	0	4	7	11
2021-ASRM no septate	0	0	952	952

**Table 5 diagnostics-14-02019-t005:** Distribution of cases according to CUME and 2021-ASRM classifications.

	2021-ASRM Septate	2021-ASRM Grey Zone	2021-ASRM No Septate	Total
CUME septate	14	9	0	23
CUME no septate	0	2	952	954

## Data Availability

Data are available upon reasonable request.

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
