# Peer review of "Prevalence of Septate Uterus in a Large Population of Women of Reproductive Age: Comparison of ASRM 2016 and 2021, ESHRE/ESGE, and CUME Diagnostic Criteria: A Prospective Study"

_diagnostics, 2024, doi:10.3390/diagnostics14182019_

Round 1

Reviewer 1 Report

Comments and Suggestions for Authors

The present paper aims to assess the prevalence of uterine septum by comparing the use of three different classification systems for uterine malformations.
The paper is well written, my only suggestion is to correct the sentence in line 60 where I believe there is an error in the phrasing of the sentence which is unclear.
The evidence that the ESHRE-ESGE classification tends to over-diagnose uterine septum is well known in the literature, and the paper confirms this and provides a good overview of the existing literature. 

Comments on the Quality of English Language

Correct the sentence in line 60

Author Response

  1. Comment: Correct the sentence in line 60
    1. Answer: Corrected

Reviewer 2 Report

Comments and Suggestions for Authors

In this prospective observational study, the authors adopted three diagnosis criteria of septate uterus to screen 977 women in reproductive age, and the incidence of septate uterus was 13.5% (132/977), 0.9% (9/977), and 2.4% (23/977) according to ESHRE-ESGE, ASRM and CUME, respectively. The authors concluded that the ESHRE-ESGE criteria may lead to the over-diagnosis of septate uterus compared to ASRM and CUME criteria. Although this study included the general population with a relatively large sample size, the study design, data interpretation, and results analysis have some major flaws and without novelty, and the clinical implication is limited. 

The controversy in the diagnosis criteria of septate uterus is due to the necessary of surgery treatment after the diagnosis. The evidence of negative impact of septate uterus still needs to be determined, although the incidence of septate uterus is reported higher in infertility women or women with previous miscarriages. The over-diagnosis of septate uterus may lead to unnecessary surgery, especially for the partial septate uterus. The published data did not support that the surgery treatment of isolated septate uterus can improve the pregnancy outcomes, possibly due to the different diagnosis criteria, the neglect of septate length and uterus volume, and other accompanied uterine abnormalities. However, the current clinical perspective is inclined to surgery treatment for septate uterus. Therefore, the urgency need in this field is to reach a general, standardized classification, based on the large sample size, homogeneous clinical study. This kind of observational study is lack of novelty and have limited clinical implication.    

As cited, the previous study conducted by LUDWIN et al. concluded that the ESHRE/ESGE criteria may mislabel healthy women with normal/arcuate uterus as septate uterus, thus leading to unnecessary surgery. In this study, the authors made a similar conclusion that 13.5% women can be diagnosed as septate uterus in a non-specified population, which is significantly higher than the reported prevalence in general population. Although the authors claimed that the population composition in this study is differed with previous LUDWIN’s study, the previous study also included a general population that account for 51% of included participants. In addition, the demographic information of included participants is limited, the participants can be asymptomatic or with classic symptoms, with or without reproductive problems, with or without assisted reproductive technology treatment, these conditions should be analyzed. 

The data in table 1: the parity, miscarriage and the termination of pregnancy were expressed as median with interquartile range, may not be appropriate, especially with the median and interquartile range of miscarriage and the termination of pregnancy were both “0”. 

The table 3 to table 5 can be integrated into one table, and the comparison between CUME and ASRM criteria was missed. 

Although the authors stated that the strength of this study is the inclusion of participants is not specialized, this non-specialized inclusion criteria has a major problem that the reproductive outcomes and long-term follow up information cannot be acquired, and the correlation between septate uterus and infertility/reproductive outcomes is more important. The over-diagnosis of ESHRE-ESGE criteria has already been accepted, therefore, the conclusion of this study is just a supplementary data. 

The images and interpretation of three-dimensional ultrasound results is not comprehensive. Some published ultrasound studies had made some progress in assisting the diagnosis of septate uterus, especially for the volume detection, this information should be considered in this study. 

The secondary objective was to analyze whether the presence of uterine fibroids, adenomyosis or intrauterine devices (IUD) could affect our ability for diagnosing septate uterus. However, in results part, the authors just listed the cases with the above-mentioned conditions, the impact of these condition on septate uterus diagnosis was not analyzed. 

The presentation of results was mostly relied on text description. 

Author Response

  1. This kind of observational study is lack of novelty and have limited clinical implication.    
    1. Thanks for this comment. With all respect, certainly there are previous studies comparing ESHRE-ESGE classification with CUME and ASRM old classification. But, to the best of our knowledge, there is no report including the new ASRM classification. If the reviewer is aware of such study we shall be happy to discuss it in our manuscript. No change made in the manuscript.

  1. As cited, the previous study conducted by LUDWIN et al. concluded that the ESHRE/ESGE criteria may mislabel healthy women with normal/arcuate uterus as septate uterus, thus leading to unnecessary surgery. In this study, the authors made a similar conclusion that 13.5% women can be diagnosed as septate uterus in a non-specified population, which is significantly higher than the reported prevalence in general population. Although the authors claimed that the population composition in this study is differed with previous LUDWIN’s study, the previous study also included a general population that account for 51% of included participants. In addition, the demographic information of included participants is limited, the participants can be asymptomatic or with classic symptoms, with or without reproductive problems, with or without assisted reproductive technology treatment, these conditions should be analyzed. 
    1. Thanks for this comment. We agree with the reviewer that we did not analyzed these data in our study. However, the primary aim of this study was not to determine the impact of diagnosis of septate uterus on reproductive health or clinical complaints. We just focused on the data of prevalence in this “general”, “mixed” population. In fact, this analysis should have been done retrospectively (with risk of bias) and the interpretation of results could have been misleading, particularly when interpreting data about clinical complaints and past reproductive history in women with uterine septum according to one classification but normal uterus with other different classification. For example, how to interpret a case of recurrent abortion in a woman who would have been diagnosed with uterine septum by ESHRE-ESGE classification but not for CUME or ASRM-2021 classification? Would that mean that recurrent abortion could be related to the septum if we choose ESHRE-ESGE but not if we choose ASRM-2021 or CUME? We think this kind of analysis is quite difficult and eve misleading with our study design. No change made in the manuscript.

  1. The data in table 1: the parity, miscarriage and the termination of pregnancy were expressed as median with interquartile range, may not be appropriate, especially with the median and interquartile range of miscarriage and the termination of pregnancy were both “0”. 

  1. We used median and IQR since these variables did not distributed normally. Statistically speaking, if a variable is not normally distributed mean and SD should not be used. Anyway, if the reviewer consider we should be happy to modify this table. No change made in the manuscript.

  1. The table 3 to table 5 can be integrated into one table, and the comparison between CUME and ASRM criteria was missed. 
    1. Thanks for this comment. Tables merged. CUME and ASRM comparison added

  1. Although the authors stated that the strength of this study is the inclusion of participants is not specialized, this non-specialized inclusion criteria has a major problem that the reproductive outcomes and long-term follow up information cannot be acquired, and the correlation between septate uterus and infertility/reproductive outcomes is more important. The over-diagnosis of ESHRE-ESGE criteria has already been accepted, therefore, the conclusion of this study is just a supplementary data. 
    1. See comment above

  1. The images and interpretation of three-dimensional ultrasound results is not comprehensive. Some published ultrasound studies had made some progress in assisting the diagnosis of septate uterus, especially for the volume detection, this information should be considered in this study. 
    1. Thanks for this interesting comment. We guess the reviewer refers to septum volume or cavity volume. We are sorry because we were not able to identify any paper analyzing this volume assessment, specifically in septate uterus. We would appreciate if the reviewer could provide us these references. In any case, we did not estimate septum or cavity volume. So, we cannot provide these data.

  1. The secondary objective was to analyze whether the presence of uterine fibroids, adenomyosis or intrauterine devices (IUD) could affect our ability for diagnosing septate uterus. However, in results part, the authors just listed the cases with the above-mentioned conditions, the impact of these condition on septate uterus diagnosis was not analyzed. 
    1. Thanks for this comment. We agree. We add some comments about this issue

  1. The presentation of results was mostly relied on text description. 
    1. This was done so in the attempt to reach the minimum 4000 words demanded by journal

Reviewer 3 Report

Comments and Suggestions for Authors

The current manuscript has major flaws, particularly in the design and methodology of the study, and is currently not suitable for publication.

1. In line 50, there are two run-on sentence. Please make sure to make the necessary grammatical modifications to avoid confusion for the reader.

2. In line 53, elaborate in one sentence on the gray zone. What do you mean by gray zone? What does it imply if people fall into the gray zone?

3. In line 63-64, what do you mean by unselected? add an "s" for high-risk populations".

4. In line 95, mention what type of institution and the specific setting (country/city).

5. Why were women aged 18-45 years old included? Adult women of reproductive age are aged between 18 and 49 years old.

6. It is not clear how data was collected. Was it collected via an EMR? a questionnaire? The section on data collection is missing for demographic variables.

7. For section 2.4, which variables were categorical? how were they defined? and which variables were continuous?

8. The methods section also failed to include recruitment strategies used to recruit subjects for the study.

9. In tables 3,4, and 5 mention that you are only listing counts (n=). It would be better to include % as well.

10. In line 321, remove your opinion from the discussion and focus on supporting your findings with studies from the literature.

11. In line 332, add "study" after reported.

Comments on the Quality of English Language

Several grammatical errors were detected across the manuscript.

Author Response

  1. In line 50, there are two run-on sentence. Please make sure to make the necessary grammatical modifications to avoid confusion for the reader.
    1. Corrected
  2. In line 53, elaborate in one sentence on the gray zone. What do you mean by gray zone? What does it imply if people fall into the gray zone?
    1. Explanation provided
  3. In line 63-64, what do you mean by unselected? add an "s" for high-risk populations".
    1. The authors of the cited study did not define what is “unselected populations”, so we cannot answer this question. We assumed they were women coming from the general female population, independently of clinical status. No change made.
  4. In line 95, mention what type of institution and the specific setting (country/city).
    1. Information added
  5. Why were women aged 18-45 years old included? Adult women of reproductive age are aged between 18 and 49 years old.
    1. Yes, we agree. However, we chose this age range. In Spain, the percentage of women who would seek pregnancy after 45 is certainly low. No change made.
  6. It is not clear how data was collected. Was it collected via an EMR? a questionnaire? The section on data collection is missing for demographic variables.
    1. This is now explained.
  7. For section 2.4, which variables were categorical? how were they defined? and which variables were continuous?
    1. This information is added
  8. The methods section also failed to include recruitment strategies used to recruit subjects for the study.
    1. This is now explained
  9. In tables 3,4, and 5 mention that you are only listing counts (n=). It would be better to include % as well.
    1. We understand this comment. However, the % could be calculated for rows or columns. Therefore, we decided to include the percentage for rows.
  10. In line 321, remove your opinion from the discussion and focus on supporting your findings with studies from the literature.
    1. Done
  11. In line 332, add "study" after reported.
    1. Done

Reviewer 4 Report

Comments and Suggestions for Authors

Dear authors,

I read your manuscript with great interest. The need for a single classification in uterine malformations is fundamental to reduce overdiagnosis and also overtreatment. Very often the ultrasound classification does not correspond to the need for surgical treatment. The manuscript data is clear and in agreement with the literature.

To improve the manuscript, I suggest:

-debating the management of these patients with respect to the presence of symptoms ( such as recurrent miscarriage, infertility) or in the absence of symptoms. 

- debating the usefulness or otherwise of a hysteroscopic investigation after an ultrasound diagnosis, especially for small indentations (5-10 mm) ( it could be useful this recent article doi: 10.31083/j.ceog5005099)

Best Regards

Author Response

  1. debating the management of these patients with respect to the presence of symptoms (such as recurrent miscarriage, infertility) or in the absence of symptoms.
    1. Thanks for this comment. We do think we already discuss this point in the Discussion (see lines 328-334). But we added a new sentence.
  2. debating the usefulness or otherwise of a hysteroscopic investigation after an ultrasound diagnosis, especially for small indentations (5-10 mm) (it could be useful this recent article doi: 10.31083/j.ceog5005099)
    1. Thanks for this comment. We have added a sentence in the Discussion and the reference suggested and an additional one.

Round 2

Reviewer 3 Report

Comments and Suggestions for Authors

We thank the authors for addressing all concerns. No additional changes are needed.